# Characterizing the Gut Microbial Communities of Native and Invasive Freshwater Bivalves after Long-Term Sample Preservation

**DOI:** 10.3390/microorganisms11102489

**Published:** 2023-10-04

**Authors:** Stephanie N. Vaughn, Carla L. Atkinson, Paul D. Johnson, Colin R. Jackson

**Affiliations:** 1Department of Biology, University of Mississippi, University, MS 38677, USA; snvaughn@go.olemiss.edu; 2Department of Biological Sciences, University of Alabama, Tuscaloosa, AL 35487, USA; carla.l.atkinson@ua.edu; 3Alabama Department of Conservation and Natural Resources, Alabama Aquatic Biodiversity Center, Marion, AL 36756, USA; paul.johnson@dcnr.alabama.gov

**Keywords:** microbiome, preservation, freshwater mussels, *Corbicula fluminea*, conservation

## Abstract

Freshwater mussels are important indicators of the overall health of their environment but have suffered declines that have been attributed to factors such as habitat degradation, a loss of fish hosts, climate change, and excessive nutrient inputs. The loss of mussel biodiversity can negatively impact freshwater ecosystems such that understanding the mussel’s gut microbiome has been identified as a priority topic for developing conservation strategies. In this study, we determine whether ethanol-stored specimens of freshwater mussels can yield representative information about their gut microbiomes such that changes in the microbiome through time could potentially be determined from museum mussel collections. A short-term preservation experiment using the invasive clam *Corbicula fluminea* was used to validate the use of ethanol as a method for storing the bivalve microbiome, and the gut microbiomes of nine native mussel species that had been preserved in ethanol for between 2 and 9 years were assessed. We show that ethanol preservation is a valid storage method for bivalve specimens in terms of maintaining an effective sequencing depth and the richness of their gut bacterial assemblages and provide further insight into the gut microbiomes of the invasive clam *C. fluminea* and nine species of native mussels. From this, we identify a “core” genus of bacteria (*Romboutsia*) that is potentially common to all freshwater bivalve species studied. These findings support the potential use of ethanol-preserved museum specimens to examine patterns in the gut microbiomes of freshwater mussels over long periods.

## 1. Introduction

Freshwater unionid mussels (Family: Unionidae) can indicate the overall health of their environment but often cannot be studied destructively because of the declines in natural mussel populations [1,2,3]. The loss of biodiversity is one of the key drivers of detrimental changes in ecosystems [4,5] and has particularly affected freshwater macroinvertebrates with low mobility rates such as unionid mussels. The biodiversity and abundance of these mussels have declined over the past 100 years, as illustrated by historical collections that show a greater diversity of mussel species in the 1960s and other eras [6]. Explanations for the declines in many mussel populations are not well documented, developed, or supported, with no consistent evidence for a single cause behind their decline [6,7]. Indeed, the dwindling of many mussel populations can be considered “enigmatic” [6] with mussel decline attributed to factors as diverse as the loss of fish hosts, habitat degradation and fragmentation [8,9], excessive inputs of nutrients into freshwater systems (e.g., chemical spills, water toxicity, and farmland run-off [10,11]), and climate change [12,13].

Freshwater mussels respond to changes in the physicochemical conditions of their environment and rely on the natural flow of rivers and fish hosts for their dispersal [2,14]. A recent review [15] listed the research topics that need to be addressed to develop efficient conservation strategies to mitigate the declines in mussel populations. At the top of the priority list (14 total topics) was understanding diet throughout a mussel’s life history. This topic relates to the importance of characterizing the gut microbiome of freshwater mussels, which is not well understood but has the potential to impact, and be influenced by, nutrient cycling within their system [16,17,18]. The gut microbiomes of other organisms include bacteria that provide beneficial, and potentially symbiotic, relationships that can increase the well-being and survivability of their host [19,20]. As global climate change continues to impact freshwater systems, some level of gut dysbiosis in native mussels may occur from stressors such as increasing temperatures and changing nutrient inputs, levels of turbidity, and flow rates [13,21]. Determining past, current, and predicted trends and the adaptations of the gut microbiome could help mitigate the decline in freshwater mussels and provide further insights to their role in nutrient cycling within their ecosystem.

The aim of this study was to determine whether ethanol-preserved specimens of freshwater mussels could yield representative information about the gut microbiome of the freshwater mussel microbiome and to determine if changes in the gut microbiome through time could be observed for specimens that have been stored long-term. Previous studies showed that the preservation of other animal specimens in ethanol allows for the characterization of the gut microbiome after various storage durations [22,23,24,25,26,27,28,29,30,31], but this has not been shown for freshwater mussels. We sampled specimens of nine species of mussels native to North America that had been propagated and cultured in hatchery pond systems and then preserved in ethanol for between 2 and 9 years. Of these species, four are currently listed as endangered (*Lampsilis virescens, Margaritifera marrianae, Theliderma cylindrica,* and *Venustaconcha trabalis*) by the U.S. Fish and Wildlife Service and one (*Hamiota altilis*) is listed as threatened. Utilizing propagated or preserved specimens from collections is likely to be the only option for sampling the microbiomes of these endangered organisms. To validate the use of ethanol in the preservation of the bivalve gut microbiome, field-collected specimens of the invasive Asian clam, *Corbicula fluminea* (*C. fluminea*; Family: Cyrenidae), were preserved in 95% ethanol and compared to those frozen at −20 °C for 60 days.

Our results show that the use of 95% ethanol is as valid a preservation method as freezing for bivalve specimens in terms of obtaining an effective sequencing depth and maintaining the richness of bacterial assemblages. We also provide further insight into the gut microbiomes of *C. fluminea* and native mussels. Being able to analyze the microbiomes of mussels that were preserved previously would allow for research to be conducted without the further collection of these endangered organisms, as well as monitoring changes in these ecosystems through time.

## 2. Materials and Methods

There were two parts to this study: an evaluation of the use of ethanol as a method of preserving freshwater bivalves for subsequent gut microbiome analysis and the characterization of the gut microbiomes of a variety of ethanol-preserved mussels collected and preserved over a historical period. The first part of this study used *C. fluminea* collected from Bear Creek (Bishop, AL, USA). *C. fluminea* individuals were collected and transported (2–3 h) to the laboratory at the University of Mississippi (Oxford, MS, USA) in coolers containing water from the collection site. Upon their arrival at the laboratory, the individuals were preserved in either 95% ethanol (n = 12) or frozen at −20 °C (n = 12). The specimens were preserved for 60 days. The second part of this study used whole individual specimens of native freshwater mussels that had been propagated and cultured in captivity at the Alabama Aquatic Biodiversity Center (AABC; Marion, AL, USA). The parents of the propagated specimens had been collected as part of mussel surveys conducted between 2012 and 2019 in streams and rivers throughout Alabama, Georgia, and Tennessee, USA (USFWS permit ES130300-5), and the specimens were used represent the F1 generation. The mussels were fed a shellfish diet (Reed Mariculture; Campbell, CA, USA) in base hatchery water where specimens were reared until they reached a length of 1 mm. The cultured mussels were then placed into pond systems with a continuous influx of groundwater (189 L per minute). The mussels were collected at approximately 14–16 months old (except for *Margaritifera marrianae*, which required 36 months). The whole mussels were preserved in 95% ethanol immediately after collection and stored at the AABC over periods ranging from 2 to 9 years (Table 1). These specimens were transported in their original glass collection jars to the University of Mississippi in 2021.

All specimens (the frozen *C. fluminea*, ethanol-preserved *C. fluminea*, and ethanol-preserved native mussels) were processed in the same way. Prior to dissection, the whole organisms were dipped in sterile water three times to remove any residual ethanol. The samples were then processed by making an incision to sever the adductor muscles and the shell was opened, exposing the organism’s gut, which was dissected and removed. The gut samples were placed in bead beating tubes containing a buffer solution (CD1) from a PowerSoil Pro kit (Qiagen, Germantown, MD, USA) for extractions.

### 2.1. DNA Extraction, Amplification, and Sequencing

DNA was extracted using the PowerSoil Pro kit, following the manufacturer’s instructions. Following the DNA extraction, a 250 bp portion of the V4 region of the bacterial 16S rRNA gene was sequenced using a dual-index 8-nucleotide barcoding approach [32]. This approach uses a single round of a PCR, reducing the risk of amplification artifacts. Following amplification, the presence of amplicons was verified using agarose gels, amplification products were standardized using SequalPrep plates (Life Technologies, Grand Island, NY, USA), and barcoded products were pooled prior to sequencing. The assembled library was spiked with 20% PhiX [33,34] and sequenced using an Illumina MiSeq at the University of Mississippi Medical Center (UMMC) Molecular and Genomics Core Facility.

The raw sequence files (fastq) were processed using the standard 16S rRNA pipeline of the DADA2 package version 1.26.0 [35] within R version 4.2.2 [36]. At least 80% of the sequences from each sample were retained following quality trimming: truncLen = c (240,160), maxN = 0, maxEE = c(2,2), and truncQ = 2. The quality profile plots were inspected to ensure the proper quality of the trimmed reads. During the merging of the reads, the sequences were trimmed further to account for any overhang (trimOverhang = TRUE), and sequences shorter than 250 base pairs (bp’s) and longer than 256 bp’s were removed. Sequences identified as potential chimeras, chloroplast, mitochondria, Archaea, and Eukarya were removed. The sequences were classified against the RDP v.18 database [37]. The final amplicon sequence variant (ASV) data were transformed into relative abundances (% sequence reads) of microbial taxa for downstream analyses.

### 2.2. Statistical Analysis

To reduce the chances of rare taxa creating sequencing artifacts and noise, singletons were removed from the data set using the “prune_taxa” function from the phyloseq package [38] in R. Alpha diversity was calculated using the “phyloseq_coverage” function from the metagMisc package [39] in R and assessed using the Inverse Simpson’s Index and the Observed Species Richness (richness based on repeatedly subsampling the rarefied number of sequences) of the ASVs. Multivariate analysis of variance (MANOVA) tests were used to specify which, if any, major bacterial taxa were significantly more or less abundant between the storage methods used for *C. fluminea*. Bray–Curtis dissimilarity matrices were used to compare structural differences in the bacterial communities by storage method for the *C. fluminea* samples. Permutational multivariate analysis (PERMANOVA) tests using Bray–Curtis distance matrices were performed using the “adonis2” function in the Vegan R package [40] to determine whether the storage method (*C. fluminea*) significantly affected the composition of the microbiome. One-way analysis of variance (ANOVA) tests were performed on the samples to determine differences in alpha diversity based on the storage method (frozen or ethanol) used for the short-term stored specimens (*C. fluminea)*.

MANOVA tests were used to specify if any major bacterial taxa were significantly more or less abundant between different durations of storage for the gut samples from the museum mussel species. Bray–Curtis dissimilarity matrices were used to compare structural differences in bacterial communities based on the duration of storage, the river of parental origin (i.e. the rivers that the mussel broods were derived from to propagate and culture the museum mussels used in this study), and mussel species for the museum mussel samples. PERMANOVAs were used to test for significant differences in the microbiome composition between gut samples based on storage duration, the river of parental origin, and mussel species. A two-way ANOVA of alpha diversity metrics was performed on the long-term-stored mussels to determine differences in mean diversity and richness based on the duration of storage, as well as within each river of parental origin and by mussel species. Non-metric multidimensional scaling (NMDS) ordinations were created using the metaMDS function in the Vegan package [40] to visualize these differences or for between durations of storage for the museum mussels. To obtain the “core” microbiome of the *C. fluminea* gut samples, museum mussel gut samples, and amongst all bivalves (*C. fluminea* and museum mussel species) in the dataset, the “prevalence” function in the microbiome package [41] was used. This function defined the “core” microbiome of the bivalve gut samples by calculating the relative abundance of ASVs that comprised greater than 1% of the total ASVs identified within each sample.

## 3. Results

Following trimming, merging, and the removal of chimeras, 84.1% of the 16S rRNA sequence reads were retained, and 69.6% of the reads were retained for the final dataset after the removal of sequences identified as chloroplasts. A total of 5850 ASVs from 968,965 unique sequences were used for downstream analyses. Rarefaction parameters were set to retain samples containing >2000 sequences. This removed three *C. fluminea* samples (two frozen samples and one ethanol-preserved sample) and three museum specimens (two *Theliderma cylindrica* (Tc12P2 and Tc12P3) and one *Hamiota altilis* (Ha19C2)).

There was no significant difference in the number of sequence reads recovered from the ethanol-preserved (mean ± standard error, 19,909 ± 4857) and frozen (13,080 ± 3189) *C. fluminea* or between the percentage of sequences retained after the initial screening (74.5 ± 2.94% for ethanol preserved, 67.9 ± 4.23% for frozen). For the native mussels preserved in ethanol long-term, the sample year had a significant effect (ANOVA; *p* < 0.01, F = 4.660) on the total bacterial sequence counts recovered in the dataset, although this did not reflect the length of time in storage, with samples preserved in 2016 yielding the most reads (48,499 ± 12,950), followed by those collected in 2012 (44,854 ± 18,886), 2015 (17,015 ± 4873), 2017 (18,320 ± 5893), and 2019 (13,354 ± 3009). Pairwise comparisons using a Tukey HSD test revealed significant differences in sequence counts recovered between the years 2016 and 2015 (*p*-adj < 0.05) and between 2016 and 2019 (*p*-adj < 0.05). The mean percentage of sequences retained after screening was also significantly different by preservation year (*p* < 0.001, F = 11.418), with 2017 retaining 94.2 ± 1.65% of the total sequences, 2012 retaining 78.9 ± 4.73%, 2019 retaining 73.2 ± 8.95%, 2016 retaining 63.5 ± 9.67%, and 2015 retaining 42.8 ± 19.8%. Pairwise comparisons showed significant differences between 2015 and 2012 (*p*-adj < 0.01), between 2017 and 2015 (*p*-adj < 0.01), between 2019 and 2015 (*p*-adj < 0.01), and between 2017 and 2016 (*p*-adj < 0.05). The mean sequence counts recovered also varied with mussel species (*p* < 0.01, F = 5.008), specifically between *P. connasaugaensis* (68,278 ± 7740) and to *A. triangulata* (5430 ± 2425), *H. altilis* (9735 ± 2189), *C. nebulosus* (12,760 ± 4727), and *V. trabalis* (25,923 ± 6074; *p*-adj < 0.05).

### 3.1. Characterization of the Gut Microbiomes of Ethanol-Preserved and Frozen C. fluminea

There were no significant differences in the overall gut microbiome composition between the *C. fluminea* samples that were preserved in ethanol and those that were frozen (PERMANOVA; Figure 1). There were also no significant differences in the alpha diversity of the gut microbiome, as reflected in the Inverse Simpson’s Index (56.7 ± 13.8 ethanol-preserved, 40.8 ± 11.8 frozen samples) or the Observed Species Richness (416 ± 55 and 352 ± 58) (*p* > 0.05 for all; ANOVA). Based on the 16S rRNA gene sequences recovered, the major bacterial phyla and subphyla found in the guts of the ethanol-preserved and frozen *C. fluminea* were the Firmicutes (49.7% and 48.9% of ethanol-preserved and frozen sequences, respectively), Planctomycetes (15.9% and 12.5%), Alphaproteobacteria (8.0% and 7.0%), Actinobacteria (6.2% and 5.4%), Gammaproteobacteria (3.9% and 6.0%), Verrucomicrobia (2.5% and 6.6%), Bacteroidetes (2.9% and 3.0%), and Betaproteobacteria (1.7% and 3.8%). Of the sequences, 5.1% sequences were unclassified for the samples preserved in ethanol compared to 3.2% for the frozen samples. None of these percentages of bacterial phyla/subphyla differed significantly between the ethanol-preserved and frozen samples (MANOVA; *p* > 0.05 for all).

At a finer taxonomic level, the most common bacterial families in the guts of *C. fluminea* were Peptostreptococcaceae (29.3% and 16.8% of the ethanol-preserved and frozen sequences, respectively), Clostridiaceae_1 (5.1% and 12.5%), Thermoguttaceae (4.4% and 4.5%), Methylocystaceae (3.8% and 2.5%), Verrucomicrobiaceae (0.6% and 3.9%), Lacipirellulaceae (2.5% and 1.9%), Bacillaceae_1 (1.7% and 2.1%), Planctomycetaceae (2.0% and 1.3%), Mycobacteriaceae (2.0% and 1.4%), Gemmataceae (1.7% and 1.0%), Bacteroidaceae (1.4% and 1.2%), Isosphaeraceae (1.4% and 1.1%), and Comamonadaceae (0.8% and 1.3%). The only bacterial family that differed in their percentage of the gut microbiome between the frozen and ethanol-preserved *C. fluminea* samples were Verrucomicrobiaceae (Phylum: Verrucomicrobia), accounting for a significantly higher percentage of the gut bacterial community of the frozen samples (3.9%) than the ethanol-preserved samples (0.6%; MANOVA; *p* < 0.05, F = 4.625).

### 3.2. Characterization of the Gut Microbiomes of Long-Term-Stored Museum Mussels 

Ten bacterial phyla/sub-phyla accounted for >90% of the 16S rRNA gene sequences recovered from the guts of the long-term-stored museum mussel specimens: Firmicutes (24.3% of sequences), Planctomycetes (15.50%), Gammaproteobacteria 8.05%), Alphaproteobacteria (8.01%), Actinobacteria (7.08%), Verrucomicrobia (6.94%), Betaproteobacteria (6.61%), Bacteroidetes (5.73%), Acidobacteria (3.92%), and Fusobacteria (2.65%), with 4.71% of sequences being unclassified (Figure 2A). For some of these phyla, there were significant differences in the percentages of the gut microbiome between mussel species: Fusobacteria (MANOVA; *p* < 0.001, F = 36.792), Betaproteobacteria (*p* < 0.01, F = 4.093), and Actinobacteria (*p* < 0.05, F = 3.753).

At the family level, the most abundant bacterial taxa characterized in the guts of the preserved mussel specimens were Clostridiaceae_1 (11.2% of sequences), Gemmataceae (8.03%), Peptostreptococcaceae (4.27%), Methylocystaceae (3.31%), Aeromonadaceae (2.79%), Fusobacteriaceae (2.58%), Bacteroidaceae (2.53%), Thermoguttaceae (2.06%), Steroidobacteraceae (1.64%), Iamiaceae (1.58%), Lachnospiraceae (1.47%), Puniceicoccaceae (1.46%), Erysipelotrichaceae (1.27%), Flavobacteriaceae (1.18%), and Methanocellaceae (1.13%; Figure 2B). Of these bacterial families, there were significant differences between host species in the relative abundance of Fusobacteriaceae (MANOVA; *p* < 0.001, F = 35.151), Methanocellaceae (*p* < 0.05, F = 3.276), Bacteroidaceae (*p* < 0.05, F = 3.187), Iamiaceae (*p* < 0.05, F = 3.013), and Erysipelotrichaceae (*p* < 0.05, F = 2.762).

There were significant differences in the overall gut community of the long-term-stored museum specimens based on mussel species (PERMANOVA; *p* < 0.01, F = 1.816) but not between the year of preservation or the river of parental origin. Within the mixed model (species, year, and river), mussel species accounted for the most variation in the gut bacterial community (45.3%), with year (8.23%) and the river of parental origin (2.88%) having much less significant effects. When species was removed from the model to account for the low replicability of each species derived from the same river of parental origin and year of preservation, the gut bacterial communities between the rivers of parental origin were significantly different (*p* < 0.01, F = 1.843) and accounted for 30.0% of the variation in the gut bacterial community. The year of preservation also showed significant differences between bacterial communities (*p* < 0.05, F = 1.501) but accounted for only 14.7% of the sample variation (Figure 3A,B).

The Inverse Simpson’s index values were not significantly different for gut bacterial communities between mussel species (Figure 4A), river of parental origin (Figure 4B), or preservation year (Figure 4C; *p* > 0.05 for all). The overall species richness (Species Observed) was not significantly affected by mussel species Figure 4D), river of parental origin (Figure 4E), or preservation year (Figure 4F).

### 3.3. Amplicon Sequence Variants

Prior to the removal of singletons (ASVs represented by just one sequence read), a total of 2798 ASVs were recovered from the *C. fluminea* dataset. Of these, 964 were found solely in the ethanol-preserved samples, 710 ASVs were found only in the frozen samples, and 1124 (40%) were recovered from both types of samples. After the removal of the singletons, there were fewer ASVs limited to each specific storage method (425 in the ethanol-preserved samples and 300 in the frozen samples) so that the ASVs found in the samples stored under both conditions now accounted for >60% (1124/1849) of the ASVs recovered. The most frequently identified ASVs in both the ethanol-preserved and frozen *C. fluminea* gut samples were classified as members of Firmicutes, Alphaproteobacteria, or Planctomycetes, and 9/10 of these most common ASVs were found in both frozen and ethanol-preserved gut samples (Table 2).

For the museum mussel specimens, there was no significant difference in the number of ASVs recovered from samples stored for different periods of time, as based on the year of preservation. Across all sample types and years, the mean (+SE) number of ASVs recovered was (469 + 29), ranging from samples collected in 2019 (382 + 77 ASVs) to 2016 (561 + 46 ASVs). There was a significant difference in the number of unique ASVs detected across the nine mussel species (ANOVA; *p* < 0.05, F = 2.639), largely driven by differences between *C. nebulosus* (from 2019) and *V. trabalis* (from 2015; *p*-adj < 0.05), with *V. trabalis* having more unique ASVs. There were two ASVs that were present in >90% of all samples when considering all bivalve species together. These were ASV8 (*Romboutsia sedimentorum*), which was present in all individual samples other than one *M. marrianae* (Mm17SA3) sample, and ASV34 (Enterobacteriaceae), which was only absent from the same *M. marrianae* sample (Mm17SA3) and a sample of *V. trabalis* (Vt15H2). The “core” microbiome, obtained from the “prevalence” function in the microbiome package [41], revealed ASV8 and ASV24 (*Turicibacter*) as the most frequently abundant (i.e., present three times, each as one of the top three derived “core” ASVs in mussel species) amongst the ASVs comprising >1% of all ASVs from each mussel gut sample. ASV14 (*Bacteroides luti*), ASV18 (*Methylocystis*), and ASV25 (Clostridiaceae_1) were also found at high frequencies, in two groups each, of the mussel gut samples (Table 3).

### 3.4. Comparison of the C. fluminea and Native Mussel Gut Microbiomes

Nine major bacterial phyla/sub-phyla accounted for >90% of sequence reads in the full bivalve dataset, with Firmicutes comprising 35.5% of sequences, followed by Planctomycetes (14.9%), Alphaproteobacteria (7.79%), Gammaproteobacteria (6.60%), Actinobacteria (6.50%), Verrucomicrobia (5.81%), Betaproteobacteria (4.86%), Bacteroidetes (4.48%), and Acidobacteria (2.39%), with 4.5% unclassified at the phylum level. There were significant differences between *C. fluminea* and native mussels (as a whole) in the relative abundance of Acidobacteria (MANOVA; *p* < 0.001, F = 20.609), Verrucomicrobia (*p* < 0.001, F = 6.295), Actinobacteria (*p* < 0.01, F = 3.823), Bacteroidetes (*p* < 0.01, F = 3.545), Firmicutes (*p* < 0.05, F = 2.730), Betaproteobacteria (*p* < 0.05, F = 2.429), and Planctomycetes (*p* < 0.05, F = 2.303), for all significant phyla except Firmicutes, which was proportionally more abundant in native mussel samples than in *C. fluminea*.

The overall gut bacterial community composition differed between *C. fluminea* and the native mussel species based on the Bray–Curtis dissimilarity (PERMANOVA; *p* < 0.001, F = 11.378; Figure 5), with dissimilarity scores between *C. fluminea* and native mussels averaging (± SE) 0.94 (± 0.005), compared to 0.69 (± 0.027) between *C. fluminea* and *C. fluminea* and 0.83 (± 0.02) between native mussels and native mussels. However, the bacterial species diversity within *C. fluminea* was not significantly different to that of native mussel species (*p* > 0.05, F = 0.0388), nor was the bacterial species richness significantly different (*p* > 0.05, F = 0.218). From the ten most abundant ASVs that each comprised >1% of each sample from the entire dataset, ASV8 (*Romboutsia sedimentorum*) and ASV 22 (*Methylocystis*) were detected in both *C. fluminea* and native mussel gut samples. ASV8 accounted for >1% of the sequence data in >90% of all bivalve samples (20/21 of *C. fluminea* specimens and 26/27 of native mussel specimens). ASV22 was also present in the majority of samples (14/21 of *C. fluminea* and 19/27 of native mussels). The remaining most abundant ASVs drove the differences in the bacterial community composition between *C. fluminea* and the native mussels (Figure 5). The ASVs that were more associated with the *C. fluminea* gut samples were ASV2 (*Paeniclostridium*), ASV7 (Clostridiaceae_1), ASV15 (Peptostreptococcaceae), ASV26 (*Lacipirellula*), and ASV 28 (*Romboutsia*). Those more associated with the native mussels were ASV11 (*Thermostilla*), ASV18 (*Methylocystis*), and ASV24 (*Turicibacter*).

## 4. Discussion

The method of sample storage (95% ethanol or freezing at −20 °C) had no significant impact on the gut microbiome of the invasive bivalve *C. fluminea*. There were no major differences in the overall composition of the gut bacterial communities, in species richness or diversity, or in taxonomic composition. At the finest level, there were some ASVs that were only detected in frozen or ethanol-preserved samples, but the majority of ASVs were found in *C. fluminea* individuals stored under both conditions. Thus, the choice of which of these preservation methods to use does not appear to have a drastic effect on the overall composition of the bacterial community in the gut of *C. fluminea*, and the variation between samples is more likely to result from individual host-to-host variations. Importantly, this suggests that ethanol preservation can be used to preserve bivalve samples for microbiome analysis, as suggested previously for aquatic insects and crustaceans [31], opening up the possibility of obtaining valid microbiome data from archived mussel specimens stored in ethanol.

The museum specimens collected and stored in 2016 and 2012 yielded the highest number of sequence reads from native freshwater mussel samples and over three times as many sequence reads as short-term-stored *C. fluminea*. While they belong to different taxa, this finding is in contrast to that of Heindler et al. [27], who found contemporary (i.e., short-term) samples of freshwater fish hindguts to yield twice as many sequence reads as museum-stored specimen (stored in the years 1901–2006). However, their study, as well as others [29,30], also found that long-term preservation in ethanol did not significantly decrease bacterial species richness in the gut samples of various invertebrate specimens. As in this study, this supports the idea that ethanol can be used to not only preserve the integrity of the specimen and tissue itself but also to preserve the diversity of bacterial species in the gut. Furthermore, the higher number of sequence reads that we observed in the 2016 and 2012 samples can be attributed to a limited number of samples from a single species, *P. connasaugaensis* (one in 2016 and two in 2012). Finding that sequence counts, bacterial species diversity, and the number of bacterial species observed were unaffected by storage time suggests that analyzing the gut microbiomes of ethanol-preserved specimens at a decadal scale is realistic and suggests exciting opportunities for investigating species-specific microbial interactions, as well as shifts in the gut microbiomes of specimens over periods that include climate change, habitat fragmentation (e.g., dams), and changes in physicochemical conditions.

An increasing number of studies have characterized the gut microbiome of freshwater mussels [3,42,43,44,45,46,47,48,49]. Many of these studies report similar findings to the current study in that the most abundant bacterial phyla in the guts of freshwater mussels are Firmicutes, Proteobacteria, and Planctomycetes [3,45,47]. These studies also show that the mussel species and river of origin can influence the gut microbial community, as was found here for the museum specimens, even when propagated and cultured in hatcheries. Characterizing the gut microbiomes of some North American freshwater mussel species poses an issue because of their conservation status. However, understanding their gut microbial communities and how they are impacted by environmental influences could lead to conservation strategies to mitigate their decline. As benthic bioturbators, freshwater mussels and *C. fluminea* play important ecosystem roles by filtering water and recycling nutrients while simultaneously ingesting free-living and particle-associated microbiota [2,3,44,50]. This ingestion may lead to changes in the host’s gut microbial community, subsequently impacting the host’s physiological functions, including digestion and susceptibility to disease [2,43,44,50,51]. Using museum specimens of endangered, protected, or even extinct mussel species for microbiome analyses broadens the range of hosts that can be studied and could lead to insights into declines in specific mussel populations. Furthermore, the microbial diversity among mussel species cultured in the same hatchery water suggests opportunities to use hatchery-cultured mussels for future studies to better understand feeding strategies, fish-host influence, and species-specific variations in the microbial communities of bivalves.

While the *C. fluminea* samples in this study were used to validate the use of ethanol as a preservation method for bivalve species, it is also important to characterize the bacterial community of this invasive clam as it is possible that the microbiome may play a role in the success of an invasive species [47,52,53]. One explanation that has been proposed for the “enigmatic” declines in mussel populations is the invasion of *C. fluminea*, which overlap in distribution [54,55] and may outcompete native mussels for food resources [6,56]. Chiarello et al. [47] revealed a lack of significant distinction between the gut bacterial communities of *C. fluminea* and freshwater native mussels collected from the same geographic location, suggesting that *C. fluminea* adjusts to local bacterial communities in the water column upon invasion. However, in this study, the gut bacterial communities of *C. fluminea* and long-term-stored freshwater mussels were found to be significantly distinct from one another. This distinction could reflect the lack of geographic overlap between the *C. fluminea* used in this study and the parental lineages of the mussels propagated in the hatchery, but it could also be due to host-species-specific physiological characteristics that should be further studied. *C. fluminea* are also known to go through periodic die-offs [6,57,58,59,60] which subsequently lead to the degradation of their tissue by microorganisms, potentially releasing harmful amounts of ammonia into the environment [58,59].

Prior characterizations of the gut microbiome of *C. fluminea* have found that one of the more prevalent bacterial taxa is *Romboutsia* [47,60], which was also common in our *C. fluminea* samples (ASV8, *Romboutsia sedimentorum*). We also found *Romboutsia* in long-term-stored native mussels, and this genus was previously identified in native mussels collected in the southeastern USA [3]. This may indicate that *Romboutsia* is an important component of the freshwater bivalve microbiome and is being selectively retained by many bivalve species, regardless of their location. *Romboutsia* are members of the Peptostreptococcaceae family within Clostridia in Firmicutes, and they show a fermentative metabolism using simple carbohydrates and amino acids [61]. The specific function of *Romboutsia* in the bivalve host is currently unknown, but initial studies on the human gut microbiome suggest that members of the genus *Romboutsia* may play a large role in gut health and metabolism [62].

Overall, our study shows that ethanol preservation can be used to preserve freshwater bivalves for gut microbiome analyses, as it preserves the diversity and overall taxonomic composition of the bacterial community. Our analysis of museum specimens suggests that analyzing the gut microbiomes of long-term ethanol-preserved mussel specimens is a realistic possibility and has the potential to enable investigations into shifts in the gut microbiome of a species over time. Our findings also reveal that the gut bacterial communities of *C. fluminea* and native mussel species frequently contain the genus *Romboutsia*, suggesting a potential role in bivalve gut health and metabolism. These results have important implications for understanding the impacts of invasive species on freshwater ecosystems and highlight the potential of using museum collections to gain insight into the decline of endangered or extinct mussel species. Moving forward, future research could focus on investigating the functional roles of *Romboutsia* in the gut microbiome of bivalve species and exploring the relationships between the gut microbiome and the success of an invasive species. Additionally, larger museum collections of native mussel species stored at longer time scales or larger collections of hatchery-raised mussels should be utilized to study the mussel gut microbiome to provide valuable information for conservation efforts and to help elucidate the drivers of declines in freshwater mussel populations.

## Figures and Tables

**Figure 1 microorganisms-11-02489-f001:**
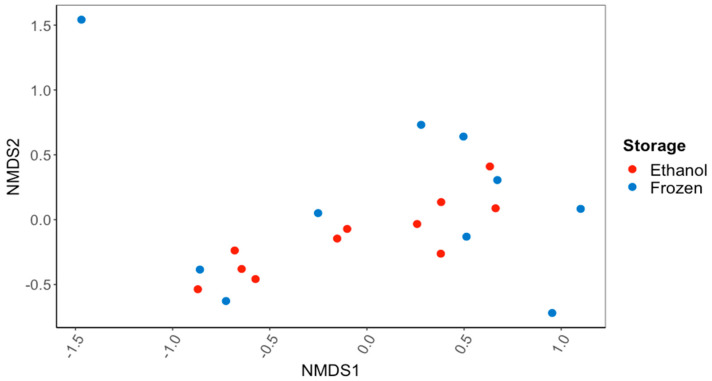
NMDS ordination based on Bray–Curtis dissimilarity scores for gut bacterial communities of the Asian Clam, *Corbicula fluminea*, preserved in 95% ethanol or frozen at −20 °C for 60 d. There was no significant difference (PERMANOVA; *p* > 0.05) in the bacterial communities recovered after the different sample preservation methods.

**Figure 2 microorganisms-11-02489-f002:**
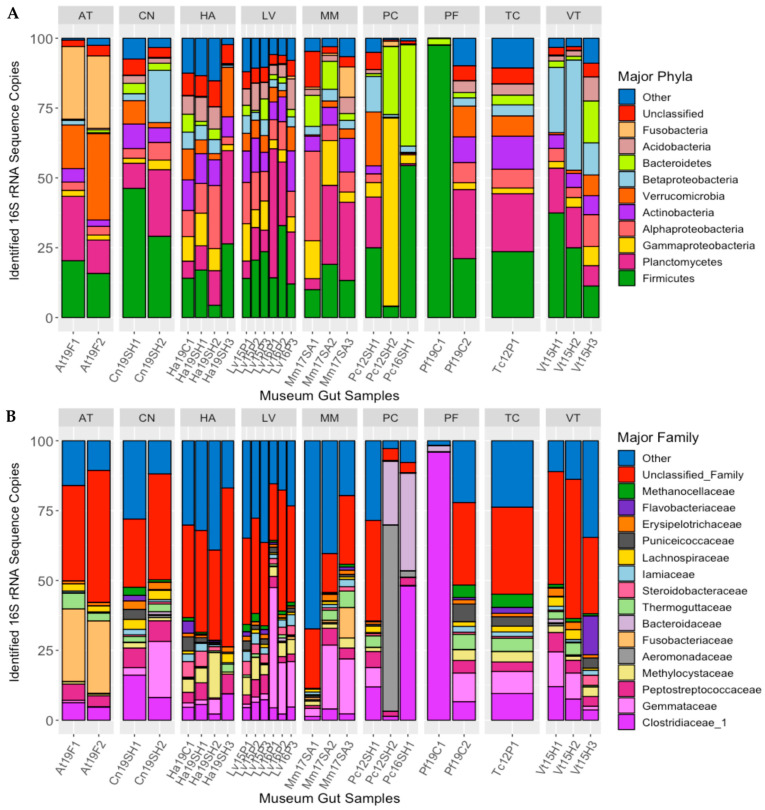
Major bacterial phyla (**A**) and families (**B**) detected in the guts of long-term stored freshwater mussels, as determined from percent of 16S rRNA gene sequences recovered. Each bar represents one individual, and bars are separated by mussel species (species abbreviations AT = *Alasmidonta triangulata* (n = 2), HA = *Hamiota altilis* (n = 4), LV = *Lampsilis virescens* (n = 6), MM = *Margaritifera marrianae* (n = 3), PC = *Pseudodontoideus connasaugaensis* (n = 3), PF = *Ptychobranchus foremanianus* (n = 2), TC = *Theliderma cylindrica* (n = 1), CN = *Cambarunio nebulosus* (n = 2), and VT = *Venustaconcha trabalis* (n = 3)). The mussels were propagated in captivity from parents of wild origin. the sample names relate to the mussel species (e.g., At = *Alasmidonta triangulata*), the last two digits of preservation year (e.g., 2019 = 19), the river of parental origin (e.g., F = Flint, SH = Shoal Creek), and the sample number within the group.

**Figure 3 microorganisms-11-02489-f003:**
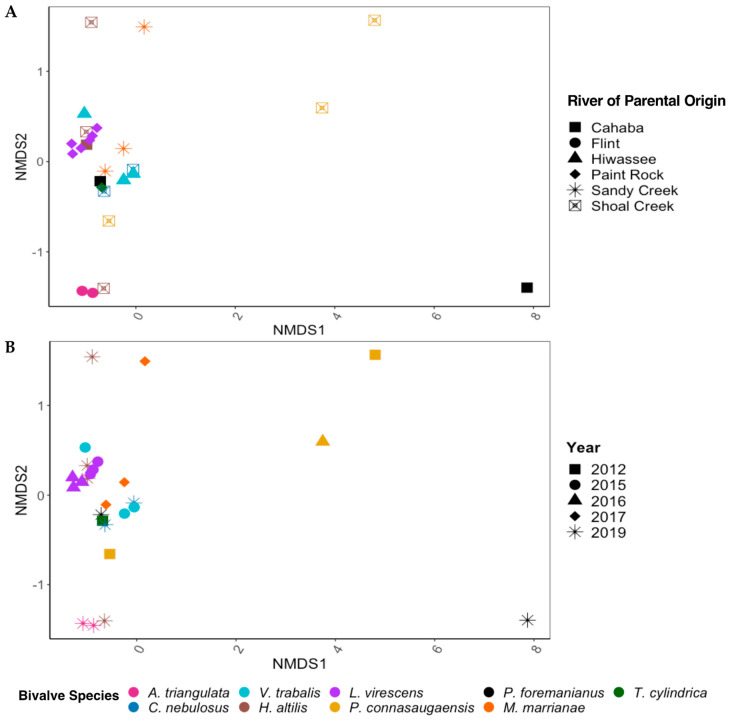
NMDS ordinations based on Bray–Curtis dissimilarity scores for the gut bacterial communities of nine species of freshwater mussels (*Alasmidonta triangulata*,* Cambarunio nebulosus*,* Venustaconcha trabalis*,* Hamiota altilis*,* Lampsilis virescens*,* Pseudodontoideus connasaugaensis*,* Ptychobranchus foremanianus*,* Margaritifera marrianae*, and *Theliderma cylindrica*) preserved in ethanol for 2–9 years. The mussels were propagated in captivity from wild parents, and ordinations are shown based on the river of parental origin (**A**) and the year of sample preservation (**B**). The gut communities were analyzed in 2021, reflecting storage durations of 2 years (2019 samples), 4 years (2017), 5 years (2016), 6 years (2015), and 9 years (2012). There were significant differences between the gut microbial communities based on mussel species (PERMANOVA; *p* < 0.01, F = 1.816) but not between the rivers of parental origin and storage duration (*p* > 0.05) within the full statistical model.

**Figure 4 microorganisms-11-02489-f004:**
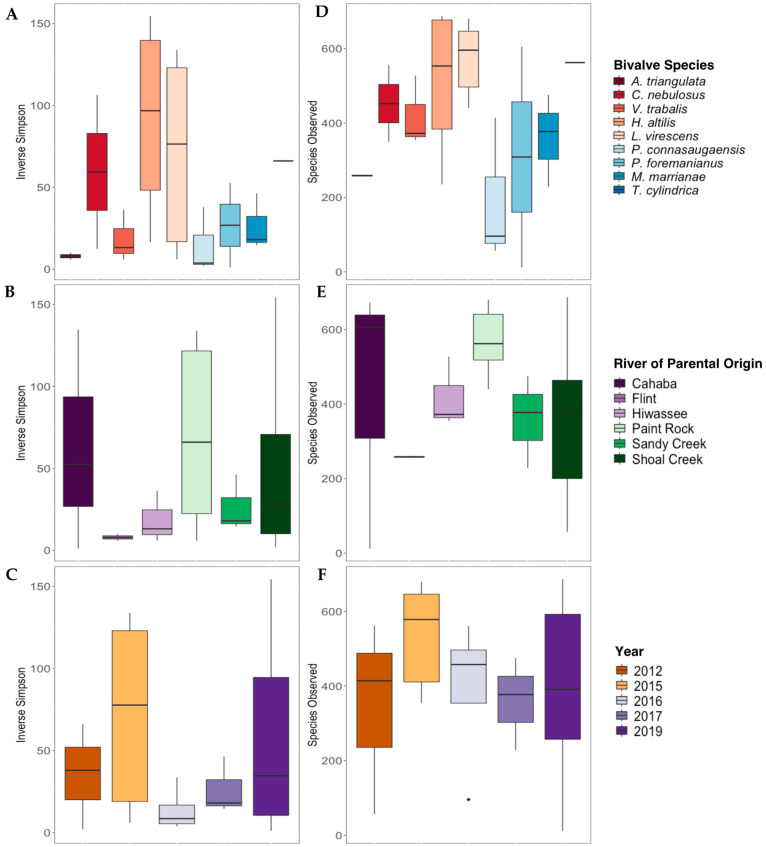
Alpha diversity metrics (Inverse Simpson’s Index, (**A**–**C**); Species Observed, (**D**–**F**)) derived from the gut bacterial communities of nine species of long-term-stored freshwater mussels. The mussels were propagated in captivity, and samples are grouped based on the mussel species ((**A**,**D**); *Alasmidonta triangulata* n = 1, *Cambarunio nebulosus* n = 2, *Venustaconcha trabalis* n = 3, *Hamiota altilis* n = 4, *Lampsilis virescens* n = 6, *Pseudodontoideus connasaugaensis* n = 3, *Ptychobranchus foremanianus* n = 2, *Margaritifera marrianae* n = 3, and *Theliderma cylindrica* n = 1), river of parental origin ((**B**,**E**); Cahaba River n = 3, Flint River n = 2, Hiwassee River n = 3, Paint Rock River n = 6, Sandy Creek n = 3, and Shoal Creek n = 7), and preservation year ((**C**,**F**); 2012 n = 3, 2015 n = 5, 2016 n = 4, 2017 n = 4, and 2019 n = 10). Boxes show the interquartile ranges/distributions of values measured in each metric, with the black solid line representing the median value from the sample type. Vertical lines represent the highest and lowest values associated with each grouping variable. Dots represent outliers from each group. There were no significant differences in the Inverse Simpson’s index values or Species Observed within species, the river of parental origin, or storage duration (*p* > 0.005).

**Figure 5 microorganisms-11-02489-f005:**
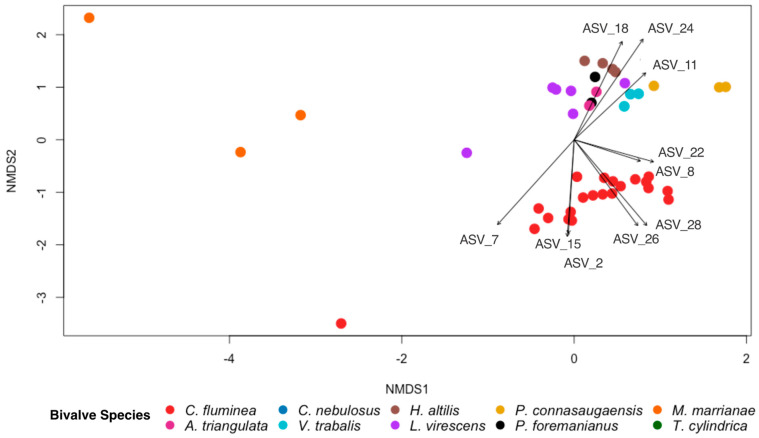
NMDS ordination based on Bray–Curtis dissimilarity scores for gut bacterial communities of the invasive Asian clam, *Corbicula fluminea*, and nine species of native North American freshwater mussels (*Alasmidonta triangulata, Cambarunio nebulosus, Venustaconcha trabalis, Hamiota altilis, Lampsilis virescens, Pseudodontoideus connasaugaensis, Ptychobranchus foremanianus, Margaritifera marrianae,* and *Theliderma cylindrica*). Ordinations are shown based on bivalve species. The ten most abundant ASVs comprising >1% of all ASVs from each sample are indicated with vector arrows for which the arrow length and direction are proportionate to their effect size and association to the samples, respectively. The gut bacterial community of *C. fluminea* was significantly different from that of the native mussel species (PERMANOVA; *p* < 0.001, F = 11.378).

**Table 1 microorganisms-11-02489-t001:** Mussels propagated and cultured in captivity for 14–16 months (except *M. marrianae*, which required 36 months) and stored in ethanol for 2–9 years after collection. Sample IDs refer to the sample names used for downstream analyses. Each species listed represents a gut sample dissected for DNA extraction and the amplification and sequencing of the V4 region of the 16S rRNA gene. Individual sample names relate to the species (e.g., At = *Alasmidonta triangulata*), the last two digits of the year of preservation (e.g., 2019 = 19), their river of parental origin (e.g., F = Flint, SH = Shoal Creek), and their sample number within their group.

Mussel Tribe	Mussel Species	Preservation Year	River of Origin	Sample ID
Alasmidontini	*Alasmidonta triangulata*	2019	Flint (AL)	At19F1
	*Alasmidonta triangulata*	2019	Flint (AL)	At19F2
	*Cambarunio nebulosus*	2019	Shoal Creek (AL)	Cn19SH1
	*Cambarunio nebulosus*	2019	Shoal Creek (AL)	Cn19SH2
	*Venustaconcha trabalis*	2015	Hiwassee (TN)	Vt15H1
	*Venustaconcha trabalis*	2015	Hiwassee (TN)	Vt15H2
	*Venustaconcha trabalis*	2015	Hiwassee (TN)	Vt15H3
Lampsilini	*Hamiota altilis*	2019	Cahaba (AL)	Ha19C1
	*Hamiota altilis*	2019	Cahaba (AL)	Ha19C2 ^1^
	*Hamiota altilis*	2019	Shoal Creek (AL)	Ha19SH1
	*Hamiota altilis*	2019	Shoal Creek (AL)	Ha19SH2
	*Hamiota altilis*	2019	Shoal Creek (AL)	Ha19SH3
	*Lampsilis virescens*	2015	Paint Rock (AL)	Lv15P1
	*Lampsilis virescens*	2015	Paint Rock (AL)	Lv15P2
	*Lampsilis virescens*	2015	Paint Rock (AL)	Lv15P3
	*Lampsilis virescens*	2016	Paint Rock (AL)	Lv16P1
	*Lampsilis virescens*	2016	Paint Rock (AL)	Lv16P2
	*Lampsilis virescens*	2016	Paint Rock (AL)	Lv16P3
	*Pseudodontoideus connasaugaensis*	2012	Shoal Creek (AL)	Pc12SH1
	*Pseudodontoideus connasaugaensis*	2012	Shoal Creek (AL)	Pc12SH2
	*Pseudodontoideus connasaugaensis*	2016	Shoal Creek (AL)	Pc16SH1
	*Ptychobranchus foremanianus*	2019	Cahaba (AL)	Pf19C1
	*Ptychobranchus foremanianus*	2019	Cahaba (AL)	Pf19C2
	*Margaritifera marrianae*	2017	Sandy Creek (AL)	Mm17SA1
Margaritiferini	*Margaritifera marrianae*	2017	Sandy Creek (AL)	Mm17SA2
	*Margaritifera marrianae*	2017	Sandy Creek (AL)	Mm17SA3
Pleurobemini	*Theliderma cylindrica*	2012	Paint Rock (AL)	Tc12P1
	*Theliderma cylindrica*	2012	Paint Rock (AL)	Tc12P2 ^1^
	*Theliderma cylindrica*	2012	Paint Rock (AL)	Tc12P3 ^1^

^1^ Indicates the removal of the sample from downstream analyses due to insufficient sequence counts.

**Table 2 microorganisms-11-02489-t002:** The core gut microbiome (most frequently identified ASVs comprising >1% of ASVs from each sample) of *C. fluminea* stored in 95% ethanol or frozen at −20 °C for 60 days. Identifications for each ASV were made to the finest classified taxonomy followed by the corresponding phylum.

Specimen	ASV	Identification	Frequency ^a^
Ethanol *C. fluminea*	ASV 8	*Romboutsia sedimentorum* (Firmicutes)	10/11
	ASV 15	Peptostreptococcaceae (Firmicutes)	9/11
	ASV 2	*Paeniclostridium* (Firmicutes)	9/11
	ASV 26	*Lacipirellula* (Planctomycetes)	8/11
	ASV 23	*Romboutsia* (Firmicutes)	8/11
	ASV 19	Clostridiales (Firmicutes)	8/11
	ASV 31	Peptostreptococcaceae (Firmicutes)	7/11
	ASV 28	*Romboutsia* (Firmicutes)	7/11
	ASV 22	*Methylocystis* (Alphaproteobacteria)	7/11
	ASV 7	*Clostridium chauvoei* (Firmicutes)	7/11
Frozen *C. fluminea*	ASV 8	*Romboutsia sedimentorum* (Firmicutes)	10/10
	ASV 15	Peptostreptococcaceae (Firmicutes)	9/10
	ASV 2	*Paeniclostridium* (Firmicutes)	9/10
	ASV 23	*Romboutsia* (Firmicutes)	8/10
	ASV 19	Clostridiales (Firmicutes)	8/10
	ASV 31	Peptostreptococcaceae (Firmicutes)	7/10
	ASV 26	*Lacipirellula* (Planctomycetes)	7/10
	ASV 22	*Methylocystis* (Alphaproteobacteria)	7/10
	ASV 7	*Clostridium chauvoei* (Firmicutes)	6/10
	ASV 39	*Methylocystis* (Alphaproteobacteria)	6/10

^a^ Frequency was determined from the number of individual gut samples that yielded that ASV.

**Table 3 microorganisms-11-02489-t003:** The most frequently identified ASVs comprising >1% of ASVs from each sample of ethanol-preserved gut samples from cultured freshwater mussels stored for 2–9 years. Identifications for each ASV were made to the finest classified taxonomy followed by the corresponding phylum.

Mussel Tribe	Mussel Species	ASV	Identification	Frequency ^a^
	*Alasmidonta triangulata*	ASV 41	Isosphaeraceae (Planctomycetes)	2/2
Alasmidontini		ASV 16	Pirellulales (Planctomycetes)	2/2
		ASV 12	Pirellulales (Planctomycetes)	2/2
	*Cambarunio nebulosus*	ASV 25	Clostridiaceae_1 (Firmicutes)	2/2
		ASV 24	*Turicibacter* (Firmicutes)	2/2
		ASV 21	Clostridiaceae_1 (Firmicutes)	2/2
	*Venustaconcha trabalis*	ASV 24	*Turicibacter* (Firmicutes)	3/3
		ASV 18	*Methylocystis* (Alphaproteobacteria)	3/3
		ASV 8	*Romboutsia sedimentorum* (Firmicutes)	3/3
Lampsilini	*Hamiota altilis*	ASV 8	*Romboutsia sedimentorum* (Firmicutes)	3/4
		ASV 24	*Turicibacter* (Firmicutes)	2/4
		ASV 720	*Planctomicrobium* (Planctomycetes)	1/4
	*Lampsilis virescens*	ASV 22	*Methylocystis* (Alphaproteobacteria)	6/6
		ASV 18	*Methylocystis* (Alphaproteobacteria)	6/6
		ASV 8	*Romboutsia sedimentorum* (Firmicutes)	6/6
	*Pseudodontoideus* *connasaugaensis*	ASV 16	Pirellulales (Planctomycetes)	3/3
		ASV 12	Pirellulales (Planctomycetes)	3/3
		ASV 8	*Romboutsia sedimentorum* (Firmicutes)	3/3
	*Ptychobranchus* *foremanianus*	ASV 14	*Bacteroides luti* (Bacteroidetes)	2/2
		ASV 10	*Aeromonas* (Gammaproteobacteria)	2/2
		ASV 965	Peptostreptococcaceae (Firmicutes)	1/2
	*Margaritifera marrianae*	ASV 4	Gemmataceae (Planctomycetes)	3/3
Margaritiferini		ASV 131	*Chryseobacterium* (Bacteroidetes)	2/3
		ASV 52	*Pseudomonas* (Gammaproteobacteria)	2/3
	*Theliderma cylindrica*	ASV 471	Selenomonadaceae (Firmicutes)	1/1
Pleurobemini		ASV 14	*Bacteroides luti* (Bacteroidetes)	1/1
		ASV 9	*Anaerobacter* (Firmicutes)	1/1

^a^ Frequency was determined from the number of individuals presenting with that ASV.

## Data Availability

Raw sequences are deposited in the NCBI Sequence Reads Archive under BioProject ID PRJNA1003077.

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
