# Peer review of "Characterizing the Gut Microbial Communities of Native and Invasive Freshwater Bivalves after Long-Term Sample Preservation"

_microorganisms, 2023, doi:10.3390/microorganisms11102489_

Round 1

Reviewer 1 Report

In this interesting manuscript, Vaughn and colleagues evaluate the feasibility of ethanol preservation as a suitable method for maintaining a reliable representation of the gut microbiome associated with freshwater bivalves over time. Considering the highly endangered status of several freshwater mussels and clams, and the pool understood factors that underlie massive mortality events, monitoring the composition of the gut microbiota of these organisms will likely cover an important role in the near future.

The manuscript is well written and easy to follow. Overall, the conclusions are scientifically sound and fully supported by the data presented by the authors, indicating that ethanol preservation is a reliable way of preserving microbial profiles for long-term storage in freshwater mussels.

I only have a few minor suggestions that could help to improve a few points of the text.

Corbicula actually does not belong to Unionida, being a member of a largely divergent superorder (Imparidentia). The authors should briefly mention the different taxonomic background of the target species and the one used for validation.  Although I don’t see any particular factor that could invalidate extending the  results of the validation experiment to Unionida, one might wonder whether the existence of linage-specific factors (e.g. microbes specifically associated with Unionida, but not with Corbicula, different tissue characteristics, etc.) may influence, to some extent, the preservation of microbial profiles in these phylogenetically distant species.

This is relevant also for section 3.4, as differences between native and invasive species might be easily explained by a number of lineage-specific traits that differentiates the physiology of unionoids and Corbicula. In other words, we might expect that a few microbial species are better fit to thrive in association with either unionoids or Corbicula in the same environment.

L97: please use metric system units, i.e. liters

L178-180 and below: please make sure to use italics for species scientific names throughout the text

L198: correct “A. 5riangulate”

Figure 3: species names should be in italics

Reviewer 2 Report

This is an interesting and important article dedicated to determining the possibility of using specimens of freshwater mussels stored in ethanol to study their gut microbiome. Authors show that ethanol preservation is a valid storage method for bivalve specimens in terms of maintaining effective sequencing depth and richness of their gut bacterial assemblages. In my opinion, the article can be published after minor revision.

1.      I recommend to authors carefully check the spelling of the Latin names of species, many names are not italicized (P.5, L. 178, 182, 197, 198, 202, 207, P.6 L. 215, 223,P. 8, L.276 and many others), or written with typing errors (for example P. 5, L. 198).

2.      Caption to Figures 1 and 3, replace Figure by Figure.

3.      Figure 2 axis signatures:

Museum gut samples, N - ?

replace «% of identified 16S sequences copies» by «identified 16S rRNA sequences copies, %»

4.      P.5, L. 174. Please, correct the sentence. Starting a sentence with a numeric value is undesirable.

5.       Please check P. 3, L. 109, P. 5, L. 167, P. 13, L. 375 - extra punctuation marks.

6.      The dates about microbial community of freshwater mussels are very important.

 In my opinion, in Discussion should include concise information about the composition of microbial communities and their role not only for C. fluminea but also for other nine species of bivalves, especially for endangered ones (Lampsilis virescens, Margaritifera marrianae, Theliderma cylindrica, Venustaconcha trabalis).

Authors should carefully check the text for typos.
